# Safety and Immunogenicity of a 23-Valent Pneumococcal Polysaccharide Vaccine (PPSV23) in Chinese Children, Adults and the Elderly: A Phase 4, Randomized, Double-Blind, Active-Controlled Clinical Trial

**DOI:** 10.3390/vaccines13080866

**Published:** 2025-08-15

**Authors:** Xiaoyu Liu, Gang Shi, Yuanyuan Dong, Wanqi Yang, Yinan Wang, Xianying Ye, Juxiang Zhang, Xinyi Yang, Dan Yu, Dan Song, Yuehong Ma, Zeng Wang, Hong Li, Weijun Hu

**Affiliations:** 1Shaanxi Provincial Center for Disease Control and Prevention, Xi’an 710054, China; liuxiaoyu594230@163.com (X.L.); 15191687306@163.com (Y.D.); nanzi519@163.com (Y.W.); 2National Institute for Food and Drug Control, Beijing 102629, China; shigang@nifdc.org.cn; 3Clinical Research Department, Sinovac Biotech Co., Ltd., Beijing 100085, China; yangwq@sinovac.com (W.Y.); yexy6494@sinovac.com (X.Y.); yud@sinovac.com (D.Y.); wangzeng@sinovac.com (Z.W.); 4Linwei Center for Disease Control and Prevention, Weinan 714009, China; liaoliao200527@163.com (J.Z.); 15591385280@163.com (D.S.); 5Clinical Research Department, Sinovac Life Sciences Co., Ltd., Beijing 102601, China; yangxy8616@sinovac.com (X.Y.); mayuehong8757@sinovac.com (Y.M.)

**Keywords:** 23-valent pneumococcal polysaccharide vaccine, safety, immunogenicity, randomized clinical trial

## Abstract

Objectives: This randomized, double-blind, active-controlled non-inferiority phase 4 clinical trial was conducted to evaluate the immunogenicity and safety of a 23-valent pneumococcal polysaccharide vaccine (PPSV23) compared to an active comparator vaccine. Methods: Pneumococcal vaccine-naïve participants aged ≥2 years were randomly assigned in a 2:1 ratio to receive a single dose of either the investigational vaccine (*n* = 1199) or the comparator vaccine (*n* = 600). Immunogenicity was evaluated at baseline and 30 days post-vaccination by measuring serotype-specific IgG antibodies against all 23 pneumococcal serotypes using enzyme-linked immunosorbent assay. The primary outcome was seroconversion, defined as a ≥two-fold increase in serotype-specific IgG antibody titers at day 30 compared to baseline. Results: At one month post-vaccination, seroconversion rates for each of the 23 serotypes ranged from 59.22% to 95.67% in the treatment group, compared to 59.66% to 94.07% in the control group. Non-inferiority was demonstrated for all serotypes, with the lower bounds of the 95% confidence intervals (95%CI) for rate differences exceeding the predefined −10% margin. Moreover, superiority was observed for 12 serotypes (6B, 23F, 1, 2, 4, 8, 9N, 9V, 11A, 15B, 17F and 18C), as the lower bounds of their 95%CI for rate differences were above 0. Adverse reactions were reported in 236 (19.68%) participants of the investigational group and 118 (19.67%) of the control group within 30 days post-vaccination, with no significant differences between groups. Conclusions: The PPSV23 vaccine administered among individual aged ≥2 years was safe, well tolerated and immunogenic, eliciting an immune response either comparable to or higher than control vaccine. These findings support its use as a safe and effective option for pneumococcal immunization.

## 1. Introduction

Pneumococcal disease caused by *Streptococcus pneumoniae* (*S. pneumoniae*, or *pneumococcus*) is a major cause of morbidity and mortality worldwide, resulting in 1.19 million deaths of lower respiratory infection in 2016 [1]. The disease disproportionately affects young children and the elderly aged 65+ years, with extensive time, place and population variations [1]. *S. pneumoniae* usually colonizes the upper respiratory tract of healthy people, with approximately 5–10% of adults and 20–60% of children carrying the bacterium in the upper respiratory tract [2]. In certain susceptible populations (e.g., people with underlying chronic conditions, the elderly aged >65 years and infants) [3], it can spread to multiple organ systems and cause invasive pneumococcal disease (IPD) (bacteremia, meningitis, osteomyelitis and septic arthritis) or noninvasive diseases (pneumonia without bacteremia, sinusitis and otitis media) [4,5]. There are at least 100 serotypes of *S. pneumoniae* [6], and the 23 serotypes contained in pneumococcal vaccines caused most IPD diseases in children and adults (approximately 60~75%) [7,8]. In China, systematic review found that IPD in children and adults was mainly caused by six serotypes, i.e., 3, 6B, 14, 19A, 19F and 23F [9,10,11]. Evaluating and monitoring the protective effect of pneumococcal vaccines routinely used in the population against different circulating serotypes after its licensure is an important measure and is essential for optimizing vaccine usage and for advocating future immunization practice and policies.

Pneumococcal vaccines are effective for the control and prevention of pneumococcal diseases [12,13]. Currently, two types of pneumococcal vaccine are available in China, i.e., the 13-valent polysaccharide conjugate vaccine (PCV13) for use in children aged 6 weeks to five years and the 23-valent pneumococcal polysaccharide vaccine (PPSV23) for use in high-risk populations aged 2 years and older [14]. However, owing to the lack of domestic evidence on disease burden, health economic data or perhaps concerns of limited vaccine supply capacity, no pneumococcal vaccines were included in China’s national immunization program, but they are administered on a private purchase basis, and vaccine uptake remains suboptimal (3–9%) [15,16].

To increase domestic access to the pneumococcal vaccine, Sinovac developed the preservative-free PPSV23, which was first approved by Chinese authorities in 2020 for use in individuals aged 2 years and older, especially in high-risk populations (i.e., the elderly, immunocompetent individuals with underlying medical conditions, immunocompromised individuals, functional or anatomic asplenia, individuals with AIDS/HIV, cerebrospinal fluid leak, and other high-risk groups). Sinovac’s PPSV23 uses advanced chromatographic techniques for polysaccharide purification to efficiently remove impurities and does not contain preservatives such as phenol. Preservative-free vaccines offer some distinct advantages, particularly for single-dose administration, making them preferable for specific populations with concerns about preservative exposure, such as young children or individuals with hypersensitivity to preservatives. Importantly, the absence of preservatives does not alter the antigenic composition or efficacy of the vaccine; rather, it aligns with global health initiatives to minimize unnecessary additives in pharmaceutical products. The pre-marketing clinical data showed that the vaccine is safe, well tolerated and effective in preventing pneumococcal disease [17]. Post-marketing surveillance also did not indicate any unanticipated safety signals. We conducted this phase 4 trial to assess the immunogenicity, tolerability, and safety of Sinovac PPSV23 compared to one internationally available PPSV23 vaccine, Pneumovax^®^ (Merck & Co., Inc., Kenilworth, NJ, USA), in children, adults and the elderly population in China. The trial was conducted in response to a request from the National Medical Products Administration (NMPA) of China and was registered with ClinicalTrials.gov (NCT05477693).

## 2. Materials and Methods

### 2.1. Study Design and Participants

This randomized, double-blind, active-controlled non-inferiority phase 4 clinical trial evaluated the immunogenicity and safety of a 23-valent pneumococcal polysaccharide vaccine (PPSV23) in the Chinese population aged ≥2 years. The study was conducted in Linwei District, Shaanxi Province, China, between 28 September 2022, and 15 April 2023. Participants were randomly assigned in a 2:1 ratio to receive either the treatment vaccine or a commercially available comparator vaccine. Written informed consent was obtained from each participant/participant’s guardian before enrollment. The trial was approved by the Ethics Committee of Shaanxi Center for Disease Control and Prevention (reference no. 2022-001-02) and was conducted in compliance with Good Clinical Practices and the ethical principles of the Declaration of Helsinki. Written informed consent was obtained from participants or children’s legal guardians prior to enrollment; adolescents aged 9–17 years also signed a written assent.

Inclusion criteria included being aged ≥2 years, in stable healthy status. The main exclusion criteria included previous history of laboratory-confirmed pneumococcal disease; receipt of a licensed or investigational pneumococcal vaccine; known history of severe allergies or reaction to vaccines or any component of vaccine; having severe neurological disease or psychosis; acute illness or axillary temperature of more than 37.0 °C; receipt of immunosuppressive therapy within the previous 6 months before enrollment, receipt of blood or blood products within the previous 3 months before enrollment; receipt of any live attenuated vaccine within the last 14 days, or any inactivated vaccine within the last 7 days. Women were excluded if they were currently pregnant, lactating or expected to be pregnant during the study. The full eligibility criteria can be found at ClinicalTrials.gov (NCT05477693).

### 2.2. Randomization and Masking

After enrollment, eligible participants within each of the three age strata (i.e., 2–17 years, 18–59 years and 60+ years) were randomly allocated to receive the treatment vaccine and the control vaccine in a 2:1 ratio. Specifically, an unmasked statistician generated the randomization list via a stratified permuted block randomization in SAS 9.4 (SAS Institute Inc., Cary, NC, USA). The unmasked study staff labeled the container of each treatment and control vaccine with a code from the randomization list. The container of treatment and control vaccine were identical in appearance, except the code number. Each participant was assigned a sequential number according to their sequence of enrollment and allocated the vaccine with the same code number on the container. During the trial, the participant, investigator assessing safety and immunogenicity outcomes and the sponsor were kept masked.

### 2.3. Procedures

The treatment vaccine is a preservative-free 23-valent PPSV (PPSV23, Sinovac Biotech Co., Ltd., Beijing, China), a sterile liquid suspension of highly purified capsular polysaccharide antigens of *S. pneumoniae* serotypes 1, 2, 3, 4, 5, 6B, 7F, 8, 9N, 9V, 10A, 11A, 12F, 14, 15B, 17F, 18C, 19A, 19F, 20, 22F, 23F and 33F. The treatment vaccine contains a mixture of 25 μg of antigens for each serotype and was packaged in a 0.5 mL prefilled syringe. The control vaccine was Pneumovax^®^ (Merck & Co., Inc., Kenilworth, NJ, USA), an internationally available PPSV23 vaccine originally manufactured in the USA and imported into China under NMPA license S20180022 [18]. The control vaccine contains the same serotypes and amounts of polysaccharide antigens as the treatment vaccine and 0.25% phenol as a preservative. The treatment and control vaccines were administered once via intramuscular injection in the deltoid muscle of the upper arm in a volume of 0.5 mL. Shipment and general storage requirements are similar for the treatment and control vaccines, requiring maintenance of a cold chain (e.g., 2–8 °C refrigeration) to preserve potency. The shelf life is 24 months.

After vaccination, all participants were monitored for immediate adverse events (AEs) for 30 min at the study site. AEs, including predefined symptoms (solicited AEs) and unspecified symptoms (unsolicited AEs), were collected on diary cards. Solicited AEs, including local AEs (e.g., injection site pain, induration, swelling, erythema, rash and pruritus) and systemic AEs (e.g., fever, acute hypersensitive reaction, cough, myalgia, arthralgia, headache and fatigue), were recorded within 7 days following vaccination, while unsolicited AEs were recorded within 30 days after vaccination. All participants (or guardians) were required to fill out a diary card and spontaneously report AEs that occurred within 30 days after vaccination. Study investigators conducted a face-to-face interview on day 7 to ensure completeness and accuracy of the safety data. Any serious adverse events (SAEs) were reported up to 30 days after enrollment. The study investigators decided the relevance of the event with vaccination. Adverse events were graded according to the guideline provided by the National Medical Product Administration [19].

Blood samples were collected for immunogenicity analysis before injection (day 0) and at day 30 after vaccination. Sera were separated immediately and stored at −20 °C until analysis. Serum IgG serotype-specific pneumococcal antibodies to 23 serotypes contained in the vaccines were tested by enzyme-linked immunosorbent assay (ELISA) at the Chinese National Institute for Food and Drug Control, Beijing, China, as described previously [20].

### 2.4. Outcomes

The primary immunogenicity outcomes were seroconversion, defined as the proportion of study participants who achieved at least a two-fold increase in serum IgG serotype-specific pneumococcal antibodies at day 30 compared with the baseline, which was found to be associated with efficacy in previous clinical trials of polyvalent pneumococcal polysaccharide vaccines [21,22]. The secondary immunogenicity outcomes included geometric mean concentration (GMC) and geometric mean fold increase (GMI) of log-transformed IgG antibodies to the 23 serotypes contained in the vaccines at day 30 post-vaccination.

The safety outcomes were based on the incidence of solicited local AEs and systemic AEs, unsolicited AEs and SAEs, including the proportion of participants reporting solicited local reactions within 7 days post-vaccination, solicited systemic reactions within 7 days post-vaccination, unsolicited AEs within 30 days post-vaccination and SAEs within 30 days post-vaccination.

### 2.5. Statistical Analysis

The sample size of 1800 was determined based on the assumed seroconversion rate of 65% in the control vaccine group for IgG antibodies to each of the six serotypes (3, 6B, 14, 19A, 19F and 23F). Assuming a non-inferiority margin of -10%, a treatment-to-control ratio of 2:1 and a one-sided significance level of 0.025, we estimated that a minimum size of 956 participants in the treatment group and 478 in the control group would be required to achieve an overall power of 80% (power for each serotype = 96.67%, calculated with the Bonferroni method). Allowing a dropout rate to 20%, the final sample was 1200 in the treatment group and 600 in the control group.

We compared log-transformed GMI relative to baseline and GMC at day 30 with Student’s *t*-test, while the seroconversion rate at day 30 was analyzed with the Chi-square test or Fisher’s exact test. We calculated the rate differences of seroconversion between the treatment and control group and its 95% confidence interval (95%CI) using unstratified analysis of the Miettinen–Nurminen method, and non-inferiority was concluded if the lower bound of the 95%CI of rate differences for each serotype in the primary outcome was larger than −10%. Adverse events were summarized descriptively as frequencies and percentages by type of event and severity. Subgroup analyses were conducted, stratifying analysis by age groups (i.e., 2–17 years, 18–59 years and 60+ years). A two-sided *p*-value of <0.05 was considered statistically significant. We conducted all analyses in SAS (version 9.4, SAS Institute Inc., Cary, NC, USA).

## 3. Results

### 3.1. Study Population

In total, 2039 participants were screened for eligibility, of whom 1800 were enrolled and randomly allocated. One participant did not receive the allocated vaccine after randomization. A total of 1199 received the treatment vaccine and 600 received the control vaccine, and they were included in the safety analysis population (*n* = 1799). In total, 1177 participants in the treatment group and 590 participants in the control group were included in the immunogenicity analysis population (*n* = 1767). A total of 32 participants were excluded because 22 had no blood draw post-vaccination, 3 participants had blood draw but outside of the visit window, 6 participants met the exclusion criteria and 1 participant in the control group received the wrong vaccine (Figure 1). Baseline characteristics, i.e., age, sex, ethnic, BMI and underlying medical conditions, were similar between the treatment and control groups (Table 1). The baseline serum IgG antibody GMCs against each of the 23 pneumococcal serotypes, ranging from 0.32 to 6.95 in the treatment group and 0.34 to 6.93 in the control group, were not significantly different at baseline (all *p* > 0.05) (Table 2), suggesting balance between the two comparison groups before vaccination.

### 3.2. Immunogenicity Post-Vaccination

After vaccination, the IgG antibodies to the 23 *S. pneumoniae* serotypes contained in both vaccines were increased significantly compared with the baseline (Table 2). The GMCs at day 30 post-vaccination ranged from 1.06 to 33.07 in the treatment group and 1.21 to 30.64 in the control group, increasing on average 3.04~12.72-fold relative to baseline in the treatment group and 2.78~10.53-fold in the control group, respectively. In both groups, serotype 14 had the highest GMCs among all the 23 serotypes assayed, while serotype 3 had the lowest. The GMC ratio between the treatment group and the control group post-vaccination ranged from 0.88 to 1.37, with serotype 3 having the lowest (0.88, 95%CI = 0.81,0.96) and serotype 4 having the highest (1.37, 95%CI = 1.25,1.49). There were nine serotypes (2, 4, 8, 9N, 9V, 11A, 15B, 18C and 23F) whose GMCs’ 95%CI had a lower bound of more than 1 (Table 2).

Regarding the seroconversion (≥2-fold increase at day 30 post-vaccination), the seroconversion rate for each of the 23 serotypes ranged from 59.22% to 95.67% in the treatment group, and in the control group from 59.66% to 94.07%, respectively. The lower bounds of the 95%CI of seroconversion rate differences (treatment group minus control group) of IgG antibodies for all 23 serotypes were all larger than −10%. Moreover, we found that there were an additional 12 serotypes (1, 2, 4, 8, 6B, 9N, 9V, 11A, 15B, 17F, 18C and 23F) which had a lower bound of the 95%CI for rate difference larger than 0 (Table 3).

### 3.3. Subgroup Analysis

Subgroup immunogenicity analysis by age group showed that GMCs of IgG antibodies against each of the 23 serotypes increased with age at baseline, with elderly aged 60+ years the highest, adults aged 18–59 years moderate and children aged 2–17 years the lowest (Appendix A). Except for a few serotypes (i.e., serotype 3 in children aged 2–17 years), the difference in seroconversion rate for most of the 23 serotypes in each age group achieved prespecified immunogenicity non-inferiority criteria (Figure 2). Wider confidence intervals were observed for seroconversion rate differences in the separate analysis, because of the small sample size and lower statistical power.

### 3.4. Adverse Reactions

In total, 236 (19.68%) participants in the treatment group and 118 (19.67%) in the control group reported adverse reactions within 30 days after vaccination. No significant differences in incidence of adverse reactions were found between the two comparison groups. The most common injection-site and systemic adverse reaction was vaccination site pain and fever. Most adverse reactions were Grade 1 and Grade 2 events in severity, but Grade 3 events were reported in eight (0.67%) participants in the treatment group and five (0.83%) in the control group (*p* = 0.7697). No deaths or Grade 4 or above adverse reactions were reported during the study period. The most commonly reported Grade 3 adverse reaction was fever (Table 4). During the study period, nine episodes of SAEs were reported by six participants, all of which were in the treatment group (Appendix A). All except one episode of SAE (urticaria with high fever occurring on day 20 after vaccination) was determined by the study investigator as related to the vaccination; the rest of the SAEs were considered unrelated.

## 4. Discussion

The World Health Assembly endorsed the Immunization Agenda 2030 (IA2030) in August 2020, setting up seven strategic priorities to expand immunization delivery across the life course, reach high equitable immunization coverage and build sustained supply and access to vaccines [23]. China has a substantial burden of pneumococcal disease [24,25]. Conducting a post-marketing study on the approved vaccines can secure safe and effective use of the product in the population, optimizing immunization practice and recommendations, as well as improving public awareness on equitable access to the vaccine. In this phase 4 trial, we evaluated the safety and immunogenicity of one PPSV23 in a wide age group and large sample. Our findings showed that the Sinovac PPSV23 vaccine is immunologically non-inferior to active control vaccine, with the lower bound of the 95%CI for seroconversion rate differences for all the 23 serotypes achieving prespecified criteria (>−10%). The non-inferiority of the tested PPSV23 vaccine to active control vaccine can be readily claimed. Moreover, there were 12 serotypes that had a lower bound of the 95%CI that was larger than 0 for seroconversion rate differences, suggesting these antigens might elicit a superior IgG antibody response.

Our immunogenicity results in this post-market study were consistent with one pivotal phase 3 clinical trial of the studied vaccine, in which the IgG antibody response was compared with another active control PPSV23 vaccine (Chengdu Institute of Biological Products Co., Ltd., Beijing, China) [17]. The seroconversion rate (≥2-fold increase post-vaccination) of the treatment vaccine for the 23 serotypes ranged from 49.71% to 90.96%, similar to this study (from 59.22% to 95.67%). In that study, all the 23 serotypes met immunogenicity non-inferiority criteria for seroconversion rate, and 10 serotypes (1, 2, 6B, 8, 9N, 9V, 15B, 17F, 18C and 23F) had a superior antibody response. These results and serotypes were repeatedly observed in the current study, suggesting a real effect was produced by Sinovac PPSV23. In an exploratory analysis stratified by age, Sinovac’s PPSV23 can elicit robust immune responses in children and adolescents, adults and elderly. The highest immune responses were observed in children and adolescents in terms of ≥2-fold increase rates for most serotypes, and the lowest were in the elderly, while GMCs showed an increasing trend with age, suggesting the pre-existing antibody levels and the immune function may influence the immune response. The results suggested that a single dose of PPV23 can provide protection against vaccine-related serotypes in individuals aged 2 years and older, especially for high-risk groups such as the elderly and immunocompromised individuals.

We found that serotype 3 induced the lowest IgG GMCs in both the treatment group (1.06, 95%CI = 1.01–1.12) and the control group (1.21, 95%CI = 1.13–1.29), which was consistent with previous studies of PPSVs and PCVs [26,27]. Structural biology investigations suggest that the thick capsular polysaccharide and a large amount of capsule shedding may underlie its inherently poor immunogenicity [28]. The clinical protection of serotype 3 is controversial. One study in Japan reported 43.1% vaccine effectiveness of PPSV23 against serotype 3 IPD among adults, and another study in the US estimated the effectiveness of PCV13 to be 79.5% against serotype 3 IPD in children aged 2 to 59 months. However, some studies found no evidence of effectiveness [29]. Consequently, serotype 3 is recognized as a shared immunological challenge for PPSV23 and all licensed PCVs, highlighting the need for next-generation vaccine strategies to prioritize its optimization.

The safety profile of the treatment vaccine was also similar to results in previous studies on PPSV23 [17,30,31]. The most common local and systemic adverse reactions were pain and fever, and most of these events occurred within 7 days and were mild in severity (i.e., Grade 1 and Grade 2 events). Although the frequency of adverse reactions varied between studies, no significant differences were observed between active controls, suggesting a variation in study population or event definition might exist between studies. One serious adverse event of urticaria was determined by the investigator to be related to the vaccination. In a review of this case, the sponsor identified insufficient evidence of a causal association with vaccination. The reason was that this event occurred a long time (20 days) after vaccination and was accompanied with fever (maximum body temperature 39.5 °C) and elevated white blood cell count, which may possibly be related to other reasons. World Health Organization guidance mentioned that severe allergic reactions (including urticaria and vascular edema, etc.) caused by immunization usually occur within a few minutes after vaccination, rarely within 2 h after vaccination, and delayed allergic reactions may occur within 48 h [32]. Further large-scale post-marketing surveillance studies focusing on urticaria would be warranted to clarify this observation.

These findings carry important policy implications. China remains one of the few countries yet to incorporate pneumococcal vaccines into its Expanded Program on Immunization (EPI). Given China’s rapidly aging population and the increasing burden of non-communicable diseases [33], integrating PPSV23 into the EPI for high-risk groups could substantially improve vaccine accessibility, enhance cost-effectiveness and mitigate coverage disparities. However, critical challenges—including sustainable financing, supply chain optimization and public health awareness campaigns—must be carefully addressed. A phased implementation strategy, initially targeting high-burden regions while leveraging existing adult vaccination infrastructure, may facilitate an efficient rollout. Such an initiative would not only strengthen China’s preventive healthcare but also provide a replicable model for other middle-income nations undergoing similar demographic and epidemiological shifts.

Our study has several limitations. First, our observation period (30 days) was too short to evaluate immune persistence. The previous studies showed that PPSV23-induced protection declines fast, with most studies having a duration of 2 to 5 years [12,34,35]. We have evaluated the immune persistence in another study, and the results showed that the antibody levels for most serotypes at 6 years remained higher than prior to vaccination in different age groups, although there was a significant decrease compared to 28 days, which suggested PPV23 may provide protection against most serotypes for at least 6 years after vaccination [36]. Second, we intentionally excluded participants with a prior history of any pneumococcal vaccine before enrollment. We were unable to evaluate the benefits of a sequential immunization schedule with PCV followed by PPSV23. Finally, the trial did not evaluate the protective effect of the test vaccine against clinical outcomes. The effectiveness of the vaccine in preventing IPD, pneumonia or acute otitis media should be further assessed in the future.

## 5. Conclusions

In summary, the tested PPSV23 vaccine, administered once among individuals aged ≥2 years and who had no prior history of pneumococcal vaccines, was safe, well tolerated and immunogenic, eliciting an immune response either comparable to or higher than the active control vaccine. The findings of this study support the use of the PPSV23 in the population aged ≥2 years.

## Figures and Tables

**Figure 1 vaccines-13-00866-f001:**
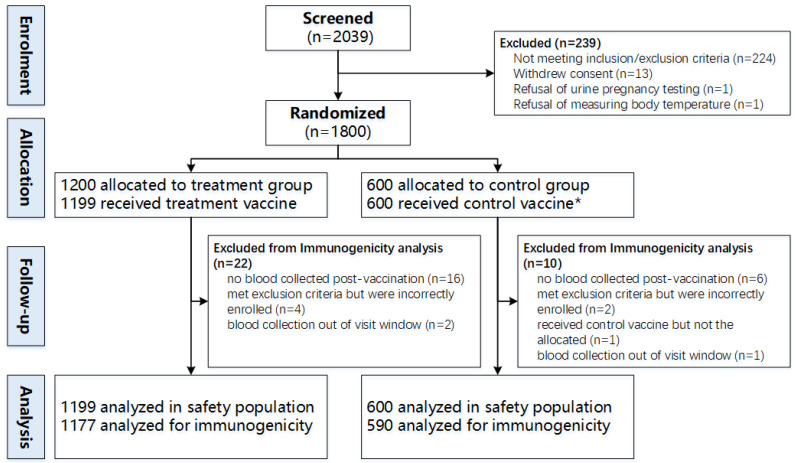
Trial profile. * One participant in the control group received the other control vaccine, not the randomly allocated one.

**Figure 2 vaccines-13-00866-f002:**
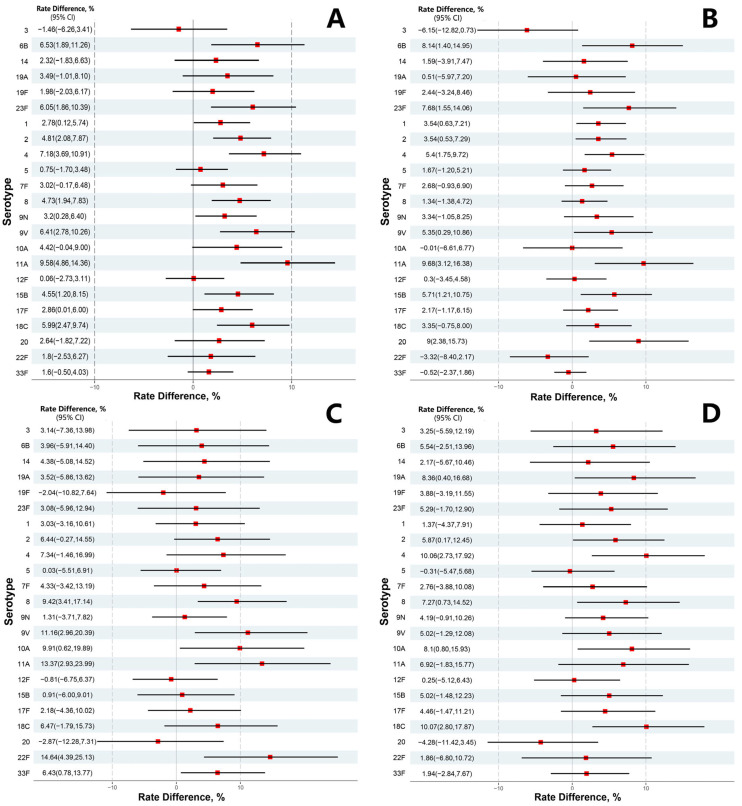
Seroconversion rate difference after vaccination between the treatment group and control group. (**A**) All participants; (**B**) children aged 2–17 years; (**C**) adults aged 18–59 years; (**D**) elderly people aged 60+ years.

**Table 1 vaccines-13-00866-t001:** Participant population demographic characteristics.

Characteristics	Safety Population, No. (%)	
	Treatment	Control
No. of participants	1199	600
Mean age in years (SD)	33.5 (25.56)	33.7 (25.50)
Age group		
2–17 years	599 (49.96)	300 (50.00)
18–59 years	240 (20.02)	121 (20.17)
60+ years	360 (30.03)	179 (29.83)
Sex		
Male	582 (48.54)	297 (49.50)
Female	617 (51.46)	303 (50.50)
Ethnicity		
Han nationality	1197 (99.83)	599 (99.83)
Others	2 (0.17)	1 (0.17)
BMI, kg/m^2^ (SD)	22.11 (4.86)	22.08 (4.66)
Underlying medical conditions	125 (10.42)	67 (11.17)

Note: SD = standard deviation; BMI = body mass index.

**Table 2 vaccines-13-00866-t002:** Serological response of study participants to the 23-valent pneumococcal polysaccharide vaccines.

Serotypes	GMC (95%CI) at Baseline		*p*-Value	GMC (95%CI) Post-Vaccination		GMC Ratio (95%CI) Post-Vaccination	*p*-Value	GMI (95%CI) Post-Vaccination		*p*-Value
	Treatment (*n* = 1177)	Control (*n* = 590)		Treatment (*n* = 1177)	Control (*n* = 590)			Treatment (*n* = 1177)	Control (*n* = 590)	
3	0.35 (0.32–0.38)	0.37 (0.34–0.42)	0.2858	1.06 (1.01–1.12)	1.21 (1.13–1.29)	0.88 (0.81–0.96)	0.0035	3.04 (2.85–3.25)	3.22 (2.93–3.54)	0.3387
6B	2.04 (1.94–2.14)	2.04 (1.91–2.19)	0.9505	6.92 (6.48–7.39)	6.45 (5.88–7.08)	1.07 (0.96–1.20)	0.227	3.39 (3.22–3.58)	3.15 (2.93–3.40)	0.1182
14	6.95 (6.64–7.27)	6.93 (6.50–7.38)	0.9435	33.07 (31.01–35.28)	30.64 (27.97–33.56)	1.08 (0.97–1.21)	0.1795	4.76 (4.47–5.07)	4.42 (4.04–4.84)	0.1873
19F	3.36 (3.22–3.51)	3.41 (3.21–3.63)	0.6912	14.23 (13.43–15.09)	12.95 (11.92–14.06)	1.10 (0.99–1.22)	0.0664	4.24 (4.02–4.46)	3.80 (3.53–4.09)	0.0171
19A	3.71 (3.54–3.90)	3.79 (3.54–4.06)	0.624	12.98 (12.26–13.75)	12.31 (11.34–13.35)	1.06 (0.96–1.17)	0.2909	3.50 (3.31–3.69)	3.25 (3.01–3.51)	0.1198
23F	1.51 (1.44–1.58)	1.46 (1.36–1.56)	0.4271	6.43 (6.05–6.84)	5.35 (4.91–5.84)	1.2 (1.08–1.34)	0.0008	4.27 (4.04–4.50)	3.67 (3.40–3.97)	0.0017
1	0.75 (0.70–0.80)	0.79 (0.71–0.86)	0.4281	6.13 (5.76–6.52)	5.91 (5.41–6.45)	1.04 (0.93–1.16)	0.4993	8.17 (7.64–8.74)	7.52 (6.84–8.26)	0.1564
2	3.42 (3.27–3.58)	3.54 (3.31–3.78)	0.4186	25.06 (23.81–26.38)	21.13 (19.65–22.71)	1.19 (1.09–1.30)	0.0002	7.32 (6.95–7.71)	5.97 (5.55–6.43)	<0.0001
4	0.43 (0.40–0.47)	0.42 (0.38–0.47)	0.7733	3.65 (3.47–3.84)	2.67 (2.48–2.87)	1.37 (1.25–1.49)	<0.0001	8.46 (7.78–9.20)	6.32 (5.61–7.11)	<0.0001
5	0.61 (0.58–0.65)	0.63 (0.58–0.68)	0.6485	4.46 (4.22–4.72)	4.11 (3.79–4.45)	1.09 (0.99–1.20)	0.0962	7.29 (6.86–7.75)	6.56 (6.02–7.15)	0.0494
7F	1.17 (1.11–1.25)	1.21 (1.11–1.32)	0.5463	7.38 (6.94–7.83)	6.77 (6.22–7.38)	1.09 (0.98–1.21)	0.11	6.28 (5.91–6.67)	5.59 (5.13–6.08)	0.027
8	2.15 (2.04–2.27)	2.16 (2.00–2.33)	0.9238	18.61 (17.72–19.54)	14.8 (13.81–15.86)	1.26 (1.15–1.37)	<0.0001	8.66 (8.14–9.22)	6.86 (6.28–7.49)	<0.0001
9N	1.14 (1.05–1.25)	1.19 (1.05–1.35)	0.6068	12.02 (11.35–12.73)	10.16 (9.37–11.02)	1.18 (1.07–1.31)	0.0009	10.5 (9.67–11.41)	8.53 (7.59–9.58)	0.0043
9V	1.49 (1.40–1.60)	1.54 (1.39–1.69)	0.6526	7.26 (6.85–7.69)	6.39 (5.88–6.94)	1.14 (1.03–1.26)	0.013	4.86 (4.59–5.14)	4.16 (3.84–4.50)	0.0018
10A	2.89 (2.77–3.02)	2.89 (2.72–3.07)	0.9676	10.97 (10.27–11.71)	9.86 (8.98–10.81)	1.11 (0.99–1.25)	0.0653	3.79 (3.60–3.99)	3.41 (3.17–3.67)	0.0197
11A	1.90 (1.77–2.03)	2.04 (1.85–2.25)	0.2446	6.67 (6.36–6.99)	5.66 (5.29–6.05)	1.18 (1.09–1.28)	<0.0001	3.51 (3.32–3.72)	2.78 (2.56–3.01)	<0.0001
12F	0.32 (0.30–0.34)	0.34 (0.31–0.37)	0.2162	2.27 (2.14–2.41)	2.25 (2.07–2.45)	1.01 (0.91–1.12)	0.8544	7.18 (6.73–7.67)	6.63 (6.04–7.27)	0.1647
15B	3.15 (2.97–3.34)	3.14 (2.89–3.41)	0.9561	20.68 (19.35–22.09)	17.06 (15.54–18.73)	1.21 (1.08–1.36)	0.001	6.56 (6.19–6.96)	5.43 (5.00–5.90)	0.0003
17F	1.25 (1.17–1.33)	1.31 (1.20–1.43)	0.3613	8.45 (7.99–8.93)	7.95 (7.35–8.60)	1.06 (0.96–1.17)	0.2192	6.77 (6.38–7.19)	6.07 (5.58–6.60)	0.0369
18C	1.26 (1.17–1.35)	1.37 (1.24–1.52)	0.1741	7.49 (7.10–7.90)	6.70 (6.20–7.23)	1.12 (1.02–1.23)	0.0189	5.96 (5.59–6.36)	4.88 (4.45–5.35)	0.0005
20	2.94 (2.78–3.11)	2.99 (2.77–3.23)	0.7141	10.00 (9.45–10.58)	10.43 (9.63–11.29)	0.96 (0.87–1.06)	0.404	3.40 (3.23–3.58)	3.48 (3.24–3.74)	0.5965
22F	2.23 (2.15–2.32)	2.24 (2.12–2.37)	0.9242	8.48 (8.08–8.90)	8.23 (7.68–8.81)	1.03 (0.95–1.12)	0.4751	3.80 (3.61–4.00)	3.67 (3.42–3.94)	0.4484
33F	1.11 (1.01–1.22)	1.20 (1.06–1.37)	0.3058	14.08 (13.18–15.03)	12.69 (11.57–13.92)	1.11 (0.99–1.24)	0.073	12.72 (11.72–13.80)	10.53 (9.38–11.82)	0.0092

Note: The values in parentheses represent 95% confidence intervals, formatted as (lower bound–upper bound). GMC = geometric mean concentration; GMI = geometric mean fold increase; 95%CI = 95% confidence interval.

**Table 3 vaccines-13-00866-t003:** Seroconversion rate of study participants inoculated with the 23-valent pneumococcal polysaccharide vaccines.

Serotypes	Treatment Group (*n* = 1177)		Control Group (*n* = 590)		Rate Difference (95%CI)	*p*-Value
	No.	Rate (95%CI)	No.	Rate (95%CI)		
3	697	59.22 (56.35–62.04)	358	60.68 (56.61–64.64)	−1.46 (−6.26–3.41)	0.5552
6B	823	69.92 (67.21–72.53)	374	63.39 (59.36–67.29)	6.53 (1.89–11.26)	0.0056
14	911	77.40 (74.90–79.76)	443	75.08 (71.39–78.53)	2.32 (−1.83–6.63)	0.2781
19F	933	79.27 (76.84–81.55)	456	77.29 (73.69–80.61)	1.98 (−2.03–6.17)	0.3382
19A	843	71.62 (68.95–74.18)	402	68.14 (64.21–71.88)	3.49 (−1.01–8.10)	0.1297
23F	931	79.10 (76.66–81.39)	431	73.05 (69.28–76.59)	6.05 (1.86–10.39)	0.0043
1	1096	93.12 (91.52–94.50)	533	90.34 (87.66–92.60)	2.78 (0.12–5.74)	0.0400
2	1104	93.80 (92.26–95.11)	525	88.98 (86.17–91.39)	4.81 (2.08–7.87)	0.0004
4	1046	88.87 (86.93–90.61)	482	81.69 (78.33–84.73)	7.18 (3.69–10.91)	<0.0001
5	1098	93.29 (91.70–94.65)	546	92.54 (90.12–94.53)	0.75 (−1.70–3.48)	0.5613
7F	1049	89.12 (87.21–90.85)	508	86.10 (83.04–88.79)	3.02 (−0.17–6.48)	0.0640
8	1099	93.37 (91.80–94.73)	523	88.64 (85.80–91.09)	4.73 (1.94–7.83)	0.0006
9N	1077	91.50 (89.76–93.03)	521	88.31 (85.43–90.79)	3.20 (0.28–6.40)	0.0311
9V	1023	86.92 (84.85–88.79)	475	80.51 (77.08–83.63)	6.41 (2.78–10.26)	0.0004
10A	860	73.07 (70.43–75.58)	405	68.64 (64.73–72.37)	4.42 (−0.04–9.00)	0.0519
11A	815	69.24 (66.52–71.87)	352	59.66 (55.58–63.65)	9.58 (4.86–14.36)	<0.0001
12F	1066	90.57 (88.75–92.18)	534	90.51 (87.85–92.75)	0.06 (−2.73–3.11)	0.9672
15B	1043	88.62 (86.66–90.37)	496	84.07 (80.86–86.93)	4.55 (1.20–8.15)	0.0072
17F	1081	91.84 (90.13–93.34)	525	88.98 (86.17–91.39)	2.86 (0.01–6.00)	0.0488
18C	1034	87.85 (85.85–89.66)	483	81.86 (78.51–84.89)	5.99 (2.47–9.74)	0.0007
20	847	71.96 (69.30–74.51)	409	69.32 (65.43–73.02)	2.64 (−1.82–7.22)	0.2483
22F	871	74.00 (71.40–76.49)	426	72.20 (68.40–75.78)	1.80 (−2.53–6.27)	0.4198
33F	1126	95.67 (94.34–96.76)	555	94.07 (91.85–95.83)	1.60 (−0.50–4.03)	0.1407

Note: The values in parentheses represent 95% confidence intervals, formatted as (lower bound–upper bound). 95%CI = 95% confidence interval.

**Table 4 vaccines-13-00866-t004:** Summary of adverse reactions in the safety analysis population.

Events	Any Severity		*p*-Value	Grade 3		*p*-Value
	Treatment Group (*n* = 1199)	Control Group (*n* = 600)		Treatment Group (*n* = 1199)	Control Group (*n* = 600)	
Overall adverse reactions within 30 days	236 (19.68)	118 (19.67)	1.0000	8 (0.67)	5 (0.83)	0.7697
Injection site adverse reactions						
Pain	191 (15.93)	98 (16.33)	0.8383	0 (0.00)	2 (0.33)	0.1111
Induration	21 (1.75)	4 (0.67)	0.0855	1 (0.08)	0 (0.00)	1.0000
Swelling	32 (2.67)	10 (1.67)	0.2458	1 (0.08)	1 (0.17)	1.0000
Erythema	20 (1.67)	6 (1.00)	0.3019	2 (0.17)	1 (0.17)	1.0000
Rash	1 (0.08)	0 (0.00)	1.0000	0 (0.00)	0 (0.00)	1.0000
Pruritus	18 (1.50)	5 (0.83)	0.2732	0 (0.00)	0 (0.00)	1.0000
Systemic adverse reactions						
Fever	32 (2.67)	18 (3.00)	0.7611	5 (0.42)	3 (0.50)	1.0000
Acute allergic reaction	2 (0.17)	1 (0.17)	1.0000	0 (0.00)	0 (0.00)	1.0000
Cough	15 (1.25)	3 (0.50)	0.2070	0 (0.00)	0 (0.00)	1.0000
Myalgia	6 (0.50)	7 (1.17)	0.1409	0 (0.00)	1 (0.17)	0.3335
Arthralgia	2 (0.17)	2 (0.33)	0.6047	0 (0.00)	0 (0.00)	1.0000
Headache	7 (0.58)	1 (0.17)	0.2818	0 (0.00)	0 (0.00)	1.0000
Fatigue	9 (0.75)	5 (0.83)	1.0000	0 (0.00)	0 (0.00)	1.0000
Dizziness	1 (0.08)	0 (0.00)	1.0000	0 (0.00)	0 (0.00)	1.0000
Pruritus	0 (0.00)	2 (0.33)	0.1111	0 (0.00)	0 (0.00)	1.0000
Urticaria	1 (0.08)	0 (0.00)	1.0000	1 (0.08)	0 (0.00)	1.0000
Limb discomfort	1 (0.08)	0 (0.00)	1.0000	0 (0.00)	0 (0.00)	1.0000
Serious adverse events (SAEs) during the study	6 (0.50)	0 (0.00)	0.1872	6 (0.50)	0 (0.00)	0.1872

Adverse events (AEs) were coded using the Medical Dictionary for Regulatory Activities, version 25.1.

## Data Availability

De-identified individual participant-level data will be available upon written request to the corresponding author following publication. Requests must be accompanied by a detailed protocol and statistical analysis plan and will be reviewed for scientific validity. For those requestors whose proposals meet the research criteria, and for which an exception does not apply, data will be transferred to requestors via a secure portal. To gain access, data requestors must enter into a data sharing agreement with Sinovac.

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
