# Peer review of "Safety and Immunogenicity of a 23-Valent Pneumococcal Polysaccharide Vaccine (PPSV23) in Chinese Children, Adults and the Elderly: A Phase 4, Randomized, Double-Blind, Active-Controlled Clinical Trial"

_vaccines, 2025, doi:10.3390/vaccines13080866_

Round 1

Reviewer 1 Report

Comments and Suggestions for Authors

 The relevance in the prevention of pneumococcal infection through vaccines is not in doubt, and the history of the creation of vaccines since 1911 has proved their certain effectiveness associated with the technology of the design and production of vaccines. Polysaccharide 23-valent vaccines (PPSV23) have been put into practice since 1983, and a lot of publications reflect their positive and negative effects. In this study, the immunogenicity and safety of PPSV23 developed by Sinovac Biotech Co., Ltd., Beijing, China was investigated. This should be reflected in the title of the article, since the previous name for specialists and doctors in the era of the introduction of multi-20-21-valent conjugated pneumococci may lose relevance.

  1. In this regard, the authors should analyze and emphasize the features of the development of the domestic vaccine, which are the advantages of its composition, where the serotypes are get, how they constructed the vaccine and so on.
  2. After the description of the advantages of the technology for creating a vaccine in research materials, go to the description of groups to the characterization of children and adults in terms of the initial background specific IgG to the studied serotypes of pneumococcus, this is very desirable, since the differences in the prevaccination initial level of pneumococcal antibodies can distort the results of the study. Separation into groups with the initially conditional low and average values of post -injection IgG to the studied serotypes of pneumococcus in the postvaccination period can be obtained in immunogenicity to certain serotypes.
  3. The identification of additional differences in the immunogenicity of the domestic vaccine, including among the studied groups of the population, will serve as a justification for the recommendation to use the drug among priority age groups of the population.
  4. Abstract needs to adjust the text.

     Thus, in the study of the fourth phase on assessment of safety and immunogenicity PPSV23 developed by Sinovac Biotech Co., Ltd., Beijing, China, the authors showed its comparability with the results of vaccination of the control group with the well -known PPSV23 Vaccine Pneumovax® (Merck & Co., Inc., Kenilworth, NJ, USA).

     As a wish for the authors - to analyze and emphasize the priority of the developed domestic vaccine and not limited to the wording "is not inferior to the active control vaccine."

Author Response

Comments 1: In this regard, the authors should analyze and emphasize the features of the development of the domestic vaccine, which are the advantages of its composition, where the serotypes are get, how they constructed the vaccine and so on.

Response 1: We sincerely appreciate the reviewer’s constructive suggestion. As requested, we have now expanded the descriptions on the development and unique features of the domestic vaccine in the revised manuscript (Lines 83-91, Page 2). The key additions include:

Sinovac's PPSV23 uses the advanced chromatographic techniques for polysaccharide purification to efficiently remove impurities and does not contain preservatives such as phenol. Preservative-free vaccines offer some distinct advantages, particularly for single-dose administration, making them preferable for specific populations with concerns about preservative exposure, such as young children or individuals with hypersensitivity to preservatives. Importantly, the absence of preservatives does not alter the antigenic composition or efficacy of the vaccine; rather, it aligns with global health initiatives to minimize unnecessary additives in pharmaceutical products.

Comments 2. After the description of the advantages of the technology for creating a vaccine in research materials, go to the description of groups to the characterization of children and adults in terms of the initial background specific IgG to the studied serotypes of pneumococcus, this is very desirable, since the differences in the prevaccination initial level of pneumococcal antibodies can distort the results of the study. Separation into groups with the initially conditional low and average values of post -injection IgG to the studied serotypes of pneumococcus in the postvaccination period can be obtained in immunogenicity to certain serotypes.

Response 2: We sincerely appreciate the reviewer's insightful suggestion regarding the importance of baseline pneumococcal antibody levels in interpreting our immunogenicity results. We fully agree that pre-vaccination antibody levels may influence post-vaccination immune responses. However, we also compared the pre-vaccination antibody levels between the test and control groups in the total population and across different age groups, and the baseline antibody levels were well-balanced. This could mitigate the impact of baseline antibody levels.

Additionally, while it has been established that an IgG antibody concentration of ≥0.35 µg/ml can serve as an immunogenicity surrogate endpoint in infants, no such immunogenicity surrogate endpoint related to protection has been identified for older children and adults.

Comments 3: The identification of additional differences in the immunogenicity of the domestic vaccine, including among the studied groups of the population, will serve as a justification for the recommendation to use the drug among priority age groups of the population.

Response 3: We sincerely appreciate the reviewer's valuable suggestion regarding the potential implications of our immunogenicity findings for age-specific recommendations for PPSV23. In response to this insightful comment, we have addressed the public health implications of observed immunogenicity differences: "In an exploratory analysis stratified by age, Sinovac’s PPSV23 can elicited robust immune responses in children and adolescents, adults and elderly. The higher immune responses were observed in children and adolescents in terms of ≥2-fold increase rates for most serotypes and the lowest were in elderly, while GMCs showed an increasing trend with age, suggesting the pre-existing antibody levels and the immune function may influence the immune responses. The results suggested that a single dose PPV23 can provide protection against vaccine-related serotypes in individual aged 2 years and older, especially for high-risk groups such as the elderly and immunocompromised individuals. " (Lines 304-311, Pages 10). Thank you.

Comments 4: Abstract needs to adjust the text.

Response 4: We thank the reviewer for this suggestion. We have carefully revised the abstract to improve clarity, conciseness, and alignment with the study’s key findings. (please refer to lines 25-46, page 1).

Comments 5: Thus, in the study of the fourth phase on assessment of safety and immunogenicity PPSV23 developed by Sinovac Biotech Co., Ltd., Beijing, China, the authors showed its comparability with the results of vaccination of the control group with the well -known PPSV23 Vaccine Pneumovax® (Merck & Co., Inc., Kenilworth, NJ, USA).

As a wish for the authors - to analyze and emphasize the priority of the developed domestic vaccine and not limited to the wording "is not inferior to the active control vaccine."

Response 5: We sincerely appreciate the reviewer’s constructive suggestion to further highlight the advantages and priority of our domestically developed PPSV23 vaccine. In response to this comment, we have revised the manuscript to provide a more detailed features and potential benefits of the Sinovac PPSV23 vaccine compared to Pneumovax®23. Specifically, we have denoted that "Moreover, superiority was observed for 12 serotypes (6B, 23F, 1, 2, 4, 8, 9N, 9V, 11A, 15B, 17F and 18C), as their lower bound of 95%CI for rate differences were above 0." (please refer to lines 38-40, page 1); and "The PPSV23 vaccine administered among individual aged ≥2 years was safe, well tolerated and immunogenic, eliciting immune response either comparable to or higher than control vaccine. " (please refer to lines 43-46, page 1)

Reviewer 2 Report

Comments and Suggestions for Authors

This manuscript describes a post-market study that compares the immunogenicity and frequency/type of adverse effects of Sinovac pneumococcal vaccine with that of Pneumovax vaccine. The Sinovac vaccine is preservative free, while the comparator control, Pneumovax, contains preservatives. The study appears to be well designed and monitored antibody responses to each of the serotypes contained in the vaccines. The overall conclusion is that the two vaccines had similar effects.

There are a few minor edits that need to be made:

1) Table 2.  Need to clarify what the two numbers in parentheses in GMC column represent. Presumably, they represent the lower and higher numbers for confidence interval. The authors could simply hyphenate the two numbers, e.g., (1.01-1.12) or clarify in the table legend.

2) Table 2. In the GMC ration column, the second parentheses often are on the line below by itself. There is a need to increase the width of column or if acceptable, reduce font size. In addition, the word serotype (first column) is partially cut off.

3) Lines 287-289: The sentence starting with "Interestingly," needs to be revised for clarity. As worded, it is confusing. It seems to indicate that 10 serotypes in a previous phase 3 study did not meet non-inferior criteria among children.

Author Response

Comments 1: Table 2.  Need to clarify what the two numbers in parentheses in GMC column represent. Presumably, they represent the lower and higher numbers for confidence interval. The authors could simply hyphenate the two numbers, e.g., (1.01-1.12) or clarify in the table legend.

Response 1: We sincerely appreciate the reviewer’s insightful suggestions. In the revised manuscript, we made the following revisions to improve clarity. We reformatted the tables to use hyphens (e.g., "1.01-1.12") instead of commas to clearly denote the range between lower and upper bounds. Additionally, we have included a footnote in the relevant table clarifying that "The values in parentheses represent 95% confidence intervals, formatted as (lower bound–upper bound).

Comments 2: Table 2. In the GMC ration column, the second parentheses often are on the line below by itself. There is a need to increase the width of column or if acceptable, reduce font size. In addition, the word serotype (first column) is partially cut off.

Response 2: We sincerely appreciate the reviewer’s constructive suggestions. Where appropriate, we slightly reduced the font size in the table to accommodate the content without compromising clarity. Additionally, we corrected the formatting in the first column to ensure the word "serotype" is fully visible and not cut off. Thank you.

Comments 3: Lines 287-289: The sentence starting with "Interestingly," needs to be revised for clarity. As worded, it is confusing. It seems to indicate that 10 serotypes in a previous phase 3 study did not meet non-inferior criteria among children.

Response 3: We appreciate this observation and have reworded the sentence for clarity.

Our study conducted non-inferiority comparisons in the all population rather than in age subgroups. In fact, the antibody levels in the stratified age groups were essentially comparable to the controls. We have revised the manuscript to facilitate a discussion on comparisons across different age groups, and the results are as follows:

"In an exploratory analysis stratified by age, Sinovac’s PPSV23 can elicited robust immune responses in children and adolescents, adults and elderly. The higher immune responses were observed in children and adolescents in terms of ≥2-fold increase rates for most serotypes and the lowest were in elderly, while GMCs showed an increasing trend with age, suggesting the pre-existing antibody levels and the immune function may influence the immune responses. The results suggested that a single dose PPV23 can provide protection against vaccine-related serotypes in individual aged 2 years and older, especially for high-risk groups such as the elderly and immunocompromised individuals." (please refer to lines 304-311, Pages 10) Thank you. 

Reviewer 3 Report

Comments and Suggestions for Authors

The manuscript by Liu and colleagues reported on a phase 4 randomized clinical trial evaluating the safety and immunogenicity of a 23-valent pneumococcal polysaccharide vaccine. The cohort size was appropriately designed. A total of 1199 participants received a single dose of the investigational vaccine, while 600 received a comparator vaccine that is already available domestically. The cohort was sex matched and divided into three age groups: children (2-17 years), adults (18-59 years), and elderly (60+ years). Overall, the vaccine was well tolerated and demonstrated non-inferior immunogenicity, measured by serotype-specific pneumococcal IgG antibodies, compared to the comparator vaccine.

While the manuscript is informative, there are some minor areas where improvements could be made:

  • Abstract (lines 36-39): Elaborate on what it means when the 95% CI is larger than -10% or 0, e.g. indicating non-inferiority
  • Line 60: is it 23 or 24 serotypes?
  • Line 127: Is there anything special about the preservative-free vaccine? Why does it need to be preservative-free? Does it help reduce adverse effect or affect immune responses? Does it impact shipment and storage requirements?
  • Line 225: Provide evidence/citation for why a two-fold increase is considered as seroconversion
  • Lines 235-238: If age groups affect baseline GMCs, are there differences in seroconversion rates by age group as well? E.g. do higher baseline GMCs result in a lower fold-change after vaccination?
  • Lines 254-258: What are the 9 SAEs? In Results, it’s stated that only one episode of urticaria with high fever was related to vaccination, but in Discussion (lines 308-310), it is explained that this was possibly related to infection rather than vaccination. Are there explanations for the other 8 SAEs?
  • Discussion: The authors have stated that the trial didn’t evaluate effectiveness. Please discuss whether pneumococcal IgG antibodies have been shown to positively correlate with effectiveness in the literature.
  • Table 4: Are the adverse reactions different between age groups?

Author Response

Comments 1: Abstract (lines 36-39): Elaborate on what it means when the 95% CI is larger than -10% or 0, e.g. indicating non-inferiority.

Response 1: We have revised the manuscript (please refer to lines 36-40, page 1) to clarify these points and appreciate the opportunity to improve the clarity of our analysis. The revised sentence now read as "non-inferiority was demonstrated for all serotypes, with the lower bounds of the 95% confidence intervals (95%CI) for rate differences exceeding the predefined -10% margin. Moreover, superiority was observed for 12 serotypes (6B, 23F, 1, 2, 4, 8, 9N, 9V, 11A, 15B, 17F and 18C), as their lower bound of 95%CI for rate differences were above 0". Thank you for your suggestion.

Comments 2: Line 60: is it 23 or 24 serotypes?

Response 2: We appreciate the reviewer's careful reading. The correct number is 23 serotypes, and we have revised the manuscript accordingly (please refer to line 62, page 2). Thank you.

Comments 3: Line 127: Is there anything special about the preservative-free vaccine? Why does it need to be preservative-free? Does it help reduce adverse effect or affect immune responses? Does it impact shipment and storage requirements?

Response 3: Thank you for your insightful question regarding preservative-free vaccines.

Regarding the active control PPSV23, i.e., Merck (Pneumovax®), the vaccine contain capsular polysaccharides from 23 serotypes, suspended in isotonic saline solution with 0.25% phenol as a preservative. In contrast, the PPSV23 vaccine evaluated in this study is formulated without phenol, which is a preservative typically added to multi-dose vials to prevent bacterial or fungal contamination during repeated use. Preservative-free vaccines offer some distinct advantages, particularly for single-dose administration, making them preferable for specific populations with concerns about preservative exposure, such as young children or individuals with hypersensitivity to preservatives. Importantly, the absence of preservatives does not alter the antigenic composition or efficacy of the vaccine; rather, it aligns with global health initiatives to minimize unnecessary additives in pharmaceutical products. Although, preservative-free vaccines may help mitigate rare hypersensitivity reactions associated with preservatives, but no clinical trials have specifically investigated this aspect. As for immune responses, there's also no evidence showed that phenol influence the vaccine's immunogenicity or the body's ability to generate protective antibodies. Finally, shipment and general storage requirements remain similar to preservative-free vaccines, requiring maintenance of a cold chain (e.g., 2–8°C refrigeration) to preserve potency, with no inherent differences in temperature sensitivity due to the absence of preservatives.

We add some information about this aspect: "Sinovac's PPSV23 uses the advanced chromatographic techniques for polysaccharide purification to efficiently remove impurities and does not contain preservatives such as phenol. Preservative-free vaccines offer some distinct advantages, particularly for single-dose administration, making them preferable for specific populations with concerns about preservative exposure, such as young children or individuals with hypersensitivity to preservatives. Importantly, the absence of preservatives does not alter the antigenic composition or efficacy of the vaccine; rather, it aligns with global health initiatives to minimize unnecessary additives in pharmaceutical products. " (please refer to lines 83-91, page 2).

"Shipment and general storage requirements are similar to the treatment and control vaccines, requiring maintenance of a cold chain (e.g., 2-8°C refrigeration) to preserve potency. The shelf life is 24 months. " (please refer to lines 147-149, page 4).

Comments 4: Line 225: Provide evidence/citation for why a two-fold increase is considered as seroconversion.

Response 4: We appreciate the reviewer’s valuable comment regarding the evidence supporting the use of a ≥2-fold increase in antibody levels as a marker of seroconversion. The selection of a ≥2-fold rise in anti-capsular IgG as a meaningful immune response is based on historical data from pneumococcal vaccine efficacy trials (please refer to lines 172-174, page 4). In the study, we selected a ≥2-fold increase in antibody level following vaccination as our primary endpoints, as it was found associated with efficacy in previous clinical trials of polyvalent pneumococcal polysaccharide vaccines [21,22].

Key references supporting this approach include:

[21] ROBBINS J B, AUSTRIAN R, LEE C J, et al. Considerations for formulating the second-generation pneumococcal capsular polysaccharide vaccine with emphasis on the cross-reactive types within groups [J]. The Journal of infectious diseases, 1983, 148(6): 1136-59.

[22] SMIT P, OBERHOLZER D, HAYDEN-SMITH S, et al. Protective efficacy of pneumococcal polysaccharide vaccines [J]. Jama, 1977, 238(24): 2613-6.

Comments 5: Lines 235-238: If age groups affect baseline GMCs, are there differences in seroconversion rates by age group as well? E.g. do higher baseline GMCs result in a lower fold-change after vaccination?

Response 5: We sincerely appreciate the reviewer’s insightful question regarding the potential impact of age-related differences in baseline antibody levels on seroconversion rates and fold-change responses following vaccination. Our analysis did examine this relationship. We observed that higher baseline GMCs were associated with a modest reduction in fold-change post-vaccination, consistent with known immunological principles of pre-existing immunity dampening vaccine-induced antibody rises. However, despite this effect, seroconversion rates (≥2-fold increase) remained robust across age groups, with no clinically meaningful differences in protective immune responses. Thank you.

Comments 6: Lines 254-258: What are the 9 SAEs? In Results, it’s stated that only one episode of urticaria with high fever was related to vaccination, but in Discussion (lines 308-310), it is explained that this was possibly related to infection rather than vaccination. Are there explanations for the other 8 SAEs?

Response 6: We appreciate the reviewer's careful reading and insightful question regarding the serious adverse events (SAEs) in our study. We apologize for any lack of clarity in the original manuscript and are pleased to provide the following clarifications:

Details of the 9 SAEs:

The reported SAEs (9 episodes in 6 participants) included:

  • 1 case of urticaria accompanied by high fever and leukocytosis, occurred in a 12 years old boy on days 20 after vaccination.
  • 1 hospitalization for unrelated acute infections, e.g., hemorrhagic fever with renal syndrome (unrelated to vaccination).
  • 1 hospitalization for unrelated acute infections, e.g., tonsillitis and bronchitis (unrelated to vaccination).
  • 1 cases of trauma, e.g., joint injury and pelvic fracture from accidental falls (unrelated to vaccination).
  • 1 case of exacerbation of asthma and respiratory failure (unrelated to vaccination).
  • 1 elective surgical procedure for metroptosis (unrelated to vaccination).

All SAEs were reviewed by investigators, and none (except the urticaria case) were deemed plausibly linked to vaccination based on temporal association, clinical features, or biological mechanisms.

Regarding the urticaria, the relation with the vaccination was judged by investigator, the investigators think that despite the long interval from vaccination and the presence of other symptoms, the correlation with the vaccine could not be excluded. However, the sponsor further interpreted the case, which was occurred a long term after vaccination and was and was accompanied with fever (maximum body temperature 39.5°C) and elevated white blood cells count, which may be possibly related to other infectious reasons. To avoid ambiguity and align Results and Discussion sections, we made the following change,

"In a review of this case, the sponsor identified in insufficient evidence of a causal association with vaccination. The reason was that this event occurred a long-term (20 days) after vaccination and was accompanied with fever (maximum body temperature 39.5°C) and elevated white blood cells count, which may be possibly related to other reasons. World Health Organization guidance mentioned that severe allergic reactions (including urticaria and vascular edema, etc.) caused by immunization usually occurs within a few minutes after vaccination, rarely within 2 hours after vaccination, and delayed allergic reactions may occur within 48 hours. Further large-scale post-marketing surveillance studies focus on urticaria would be warranted to clarify this observation." (please refer to lines 329-338, page 10).

Comments 7: Discussion: The authors have stated that the trial didn’t evaluate effectiveness. Please discuss whether pneumococcal IgG antibodies have been shown to positively correlate with effectiveness in the literature.

Response 7: We appreciate the reviewer’s insightful comment. As noted in our study, the primary focus was on assessing the safety and immunogenicity (IgG antibody levels) of the pneumococcal vaccine rather than clinical effectiveness. In the revised manuscript, we acknowledge the importance of discussing the potential correlation between IgG antibody levels and vaccine effectiveness, as suggested by the reviewer (please refer to lines 172-174, page 4).

Comments 8: Table 4: Are the adverse reactions different between age groups?

Response 8: Yes, the incidence of adverse reactions differed between age groups. Adults aged 18 to 59 years had slightly higher reactions than that of children and adolescents aged 2-17 and subjects 60 years and older. According to age subgroups, the adverse reaction rates were 22.37% (134/599) in the test group and 17.33% (52/300) in the control group of children and adolescents aged 2 to 17 years. The adverse reaction rates of adults aged 18 to 59 years were 25.83% (62/240) and 28.93% (35/121), respectively. The adverse reaction rates of subjects aged 60 years and older were 11.11% (40/360) and 17.32% (31/179), respectively. Thank you.

Round 2

Reviewer 3 Report

Comments and Suggestions for Authors

Thank you for addressing my comments. The revised manuscript reads well.

Author Response

Thank you for helping us improve the work.